# Risk Factors for Anal Continence Impairment Following a Second Delivery after a First Traumatic Delivery: A Prospective Cohort Study

**DOI:** 10.3390/jcm12041531

**Published:** 2023-02-15

**Authors:** Gabriel Marcellier, Axelle Dupont, Agnes Bourgeois-Moine, Arnaud Le Tohic, Celine De Carne-Carnavalet, Olivier Poujade, Guillaume Girard, Amélie Benbara, Laurent Mandelbrot, Laurent Abramowitz

**Affiliations:** 1Proctology and Gastroenterology Department, APHP-Bichat Hospital, 75018 Paris, France; 2Biostatistics and Medical IT Department, APHP-Bichat Hospital, 75018 Paris, France; 3Department of Obstetrics and Gynecology, APHP-Bichat Hospital, 75018 Paris, France; 4Department of Obstetrics and Gynecology, Versailles Hospital, 78157 Le Chesnay, France; 5Department of Obstetrics and Gynecology, APHP-Armand Trousseau Hospital, 75012 Paris, France; 6Department of Obstetrics and Gynecology, APHP-Beaujon Hospital, 92110 Clichy, France; 7Department of Obstetrics and Gynecology, APHP-Jean Verdier Hospital, 93140 Bondy, France; 8Department of Obstetrics and Gynecology, APHP-Louis Mourier Hospital, 92700 Colombes, France; 9Proctology and Gastroenterology Department, Blomet Clinic (Ramsay GDS Group), 75015 Paris, France

**Keywords:** anal incontinence, postpartum, endoanal sonography, OASI (obstetrical anal sphincter injury)

## Abstract

Postpartum anal incontinence is common. After a first delivery (D1) with perineal trauma, follow-up is advised to reduce the risk of anal incontinence. Endoanal sonography (EAS) may be considered to evaluate the sphincter and in case of sphincter lesions to discuss cesarean section for the second delivery (D2). Our objective was to study the risk factors for anal continence impairment following D2. Women with a history of traumatic D1 were followed before and 6 months after D2. Continence was measured using the Vaizey score. An increase ≥2 points after D2 defined a significant deterioration. A total of 312 women were followed and 67 (21%) had worse anal continence after D2. The main risk factors for this deterioration were the presence of urinary incontinence and the combined use of instruments and episiotomy during D2 (OR 5.12, 95% CI 1.22–21.5). After D1, 192 women (61.5%) had a sphincter rupture revealed by EAS, whereas it was diagnosed clinically in only 48 (15.7%). However, neither clinically undiagnosed ruptures nor severe ruptures were associated with an increased risk of continence deterioration after D2, and cesarean section did not protect against it. One woman out of five in this population had anal continence impairment after D2. The main risk factor was instrumental delivery. Caesarean section was not protective. Although EAS enabled the diagnosis of clinically-missed sphincter ruptures, these were not associated with continence impairment. Anal incontinence should be systematically screened in patients presenting urinary incontinence after D2 as they are frequently associated.

## 1. Introduction

Anal incontinence (AI) is a taboo condition and a frequent cause of handicap, present in up to 20% of the population [1,2]. Its detection requires a dedicated investigation and can be difficult [3]. Continence is a complex mechanism involving the anal sphincter, rectal compliance, anorectal angulation, pudendal nerve innervation and the nature of the stools. Incontinence occurs when one or more of these mechanisms is altered beyond compensation [4]. Aging leads to a decrease in muscle and perineal trophicity and is one of the main risk factors for AI [5]. Various events in perineal life will accelerate this aging. Childbirth is one of these disruptors of pelvic integrity. The damage it causes to the perineum (mostly anal sphincter injury and pudendal nerve stretching) is frequent, affecting 12 to 28% of parturients [6,7] and can lead to AI in 4 to 40% of women giving birth [6,7,8,9,10]. Repeated deliveries lead to a risk of cumulative damages [7,11,12,13] and obstetrical life events contribute to a higher risk of AI in women compared to men [5,14,15].

Assessment of obstetrical anal sphincter injuries (OASI) is carried out immediately after delivery by the obstetrician/gynecologist, who describes any extension of a perineal tear to the anal sphincter. However, ruptures identified during an endoanal sonography (EAS) are undiagnosed at delivery in up to 30% of cases, and may be at risk of AI [6,9]. The extension of the sphincter damage is also estimated by EAS. Despite its potential advantages, the benefits of performing this procedure are debated because the diagnosis of a sphincter tear in EAS (EAS rupture) will not always have an impact on patient care (8).

Many studies in the literature focus on AI in primiparous women, but few address the risk factors of AI after a second delivery (D2), despite the fact that the fertility rate in France is close to two and that second child deliveries approach 250,000 annually [16,17]. An even smaller proportion of these studies have been conducted prospectively and none on large cohorts [18,19,20,21]. Moreover, there is a need to further investigate the risk factors of AI after D2 among women who have become vulnerable after a first traumatic delivery (D1), since a proctologist (surgeon or gastroenterologist) is then more likely to be consulted to advise about the risks and modalities of a new delivery.

The objective of our study was to determine the risk factors of continence impairment after D2 among women who had a traumatic D1, in particular the impact of EAS ruptures.

## 2. Materials and Methods

### 2.1. Population

Our work is an ancillary study of the prospective, randomized, multicenter “Prevention of Anal Incontinence by Caesarean Section” (EPIC) study, which compared the benefit of prophylactic caesarean section (CS) to vaginal delivery (VD) at D2 in women with a history of a traumatic D1 with sphincter rupture confirmed by EAS [8]. Women were recruited in six maternity units in the Paris area between April 2008 and December 2014. They were included by their gynecologist during the 3rd trimester of their 2nd pregnancy if they met the inclusion criteria, which were as follows: a single history of traumatic vaginal delivery (VD), defined as forceps extraction or with grade III perineal tear (reaching the sphincters), age above 18 years old, informed written consent and no AI at inclusion, based on a YES/NO answer to the question asked by the gynecologist. They were excluded if they had a history of grade IV perineum tear (which corresponds to the most severe grade of OASI with sphincter and anal mucosa damage) and if a CS was indicated for their future delivery for a non-proctological reason. After inclusion, systematic prospective follow-up was carried out, with a proctological examination during the 3rd trimester of their 2nd pregnancy (referred to as before D2 visit) and then 6 months after the second delivery (referred to as after D2 visit). This visit included questionnaires measuring the Vaizey score for anal continence and the measure of urinary handicap score (MHU) for urinary continence, as well as an EAS.

In the EPIC study, women with EAS sphincter rupture were randomized to perform D2 by VD or CS. Some women included in EPIC were not randomized, either because EAS did not reveal a sphincter rupture or because they refused randomization. The mode of delivery was then discussed between the obstetrician and the patient. In this study, we included all women who were explored by EAS before D2 and for whom a Vaizey score was calculated before and after D2, regardless of their randomization status.

### 2.2. Objective and Thresholds

The analysis of anal continence was based on the Vaizey score [22] (Appendix A). Data differs in the literature to assess which Vaizey value significantly defines incontinence [23]. In the EPIC study, based on this literature and on expert opinion, a score ≥5 defined AI [8,24]. Our population being inhomogeneous regarding continence before D2, we selected as the primary endpoint worsening of the Vaizey score after D2, defined as an increase ≥2 points in the score between the two proctological examinations. Comparable definitions were used in previous proctologic studies [25]. 

Because transient AI (lasting less than 2 months) is common in the immediate postpartum period [26,27], the assessment 6 months after D2 was used to measure persistent continence deterioration. 

EAS was performed by a single trained operator, using a rotating rectal probe (7–10 MHz, Brüel and Kjaer). Upper, middle and lower anal canal were studied. A sphincter lesion was identified as a loss of continuity visible by a change in echogenicity within the sphincter ring [28]. Severity was assessed based on the Starck score (Appendix B). A score ≥9 was used to define a severe sphincter rupture [29,30]. The clinical description of perineal lesions was based on the Royal College of Obstetricians and Gynecologists classification, where the anal sphincter is considered impaired in grades III and IV (Appendix C). We defined a “hidden sphincter rupture” as a tear undiagnosed in the delivery room (or under-diagnosed as a grade I or II) but observed by EAS. After D2, ruptures were considered “de novo” if no EAS defect was visible after D1.

The analysis of urinary continence was based on the MHU score (Appendix D) [31], treated as a continuous variable ranging from 0 to 28 points. Macrosomia was defined by birthweight >4 kg [32]. Birthweight was not collected in D2 in the case of CS. Instrumental delivery referred to the use of all types of forceps or vacuum but the type of forceps was not specified. Details of the episiotomy were not collected. We defined “abnormal transit” as the presence of diarrhea, constipation or dyschesia. We asked the patients whether or not they had undergone perineal rehabilitation, but the modalities were not collected (number of sessions or technique used).

### 2.3. Statistical Analysis

Categorical variables were described as numbers and percentages and quantitative variables were described as median and interquartile ranges. We compared median Vaizey scores at the two visits with Wilcoxon paired tests. To assess the association between the primary outcome and the characteristics of the women, univariate logistic regressions were performed to determine unadjusted odds ratios (OR) and their 95% confidence intervals. Variables with a univariate *p* value < 0.20 were tested in multivariate models. Variable selection for the final multivariate model was performed using top-down selection with the Akaike information criterion. The linearity assumption was tested graphically and with the Wald test for the MHU score. An analysis on the subgroup of women giving birth by VD at D2 was conducted using the same methodology, to study the impact of a second vaginal delivery and its characteristics. *p* values < 0.05 were considered significant. The tests were two-sided. All analyses were performed using R software (v.3.4).

## 3. Results

### 3.1. Participants

A total of 549 parturients were included in the EPIC study, of which 312 had a Vaizey score completed before and after D2 and were included in our ancillary work (Figure 1). Characteristics of the population are described in Table 1.

### 3.2. Characteristics of the Deliveries

#### 3.2.1. First Delivery (D1)

Among the 312 women included, 285 (92.2%) delivered with forceps at D1 and 266 (86.9%) had episiotomies. After D1, there were 27 (8.9%) grade I, 15 (4.9%) grade II and 48 (15.7%) grade III perineal tears. An OASI was therefore only clinically visible in these 48 women. However, the EAS performed before D2 showed sphincter tears in 192 (61.5%) of these women. After D1, 225 (74.3%) women had perineal rehabilitation.

#### 3.2.2. Second Delivery (D2)

A total of 103 women (34.8%) delivered by CS and 193 (65.2%) by VD. For 16 women (5.1%), the mode of delivery was not recorded. Among the 193 VD, there were 13 forceps (6.7%), 12 vacuum (6.2%), 49 episiotomies (25.4%) and 87 grade I (45.1%), 24 grade II (12.4%) and 2 grade III (1.0%) perineal tears. After D2, 201 (64.4%) women underwent perineal rehabilitation. Characteristics of the second delivery among the subgroup of women delivering vaginally is described in Table 2.

### 3.3. Characteristics of Continence

Before D2, 43 (13.8%) women had a Vaizey score ≥ 5 and the median Vaizey score was 1 [0–3]. After D2, the median Vaizey score remained unchanged across the total population, with no significant worsening (*p* = 0.33). However, 67 women (21.5%) significantly worsened their continence score (≥2 points) and increased their median Vaizey score from 1 [0–2] before D2 to 5 [3–7] afterwards. The median MHU score was 4/28 [2–8] during the 3rd trimester of the 2nd pregnancy and 1/28 [0–4] 6 months after D2. Details of the outcomes measured 6 months after D2 are presented in Table 3.

### 3.4. Risk Factors for Deterioration of Anal Continence 6 Months after D2

#### 3.4.1. Among the 312 Included Patients

Maternal weight, ethnicity, fetal presentation, macrosomia at D1, transient AI after D1, lack of perineal rehabilitation and the presence of a sphincter rupture diagnosed in EAS before D2 (whether occult or not and whether the EAS rupture was severe or not), were not associated with an increased Vaizey score after D2 (Table 4) in univariate analysis.

In multivariate analysis, delivering by CS did not impact significantly the risk of worsening continence after D2 compared to VD (Table 5). Women who were already incontinent before D2 were less likely to deteriorate in their Vaizey score after D2 (OR 0.27–95% CI 0.08–0.86). Increase in MHU score after D2 was linearly associated with a risk of continence deterioration (OR increased by 1.24 per MHU point-95% CI 1.08–1.42).

#### 3.4.2. Among the 193 Patients That Delivered Vaginally at D2

In this subgroup, in univariate analysis, macrosomia, the absence of perineal rehabilitation after D1 or D2 and the presence of an endosonographic or clinical sphincter rupture were not associated with an increased Vaizey score after D2 (Table 6). Instrumental delivery, with or without episiotomy, increased the risk of worsening anal continence after D2. We did not find a significant association with episiotomy itself.

In multivariate analysis (Table 7), instrumental delivery or episiotomy were associated with a deterioration of anal continence only when performed together (OR 4.18–95% CI 1.05–16.59). Increase in MHU score after D2 was linearly associated with a risk of AI (OR1.25 per MHU point-95% CI 1.11–1.43).

### 3.5. Sphincter Tears

#### 3.5.1. Sphincter Tears before D2

192 (61.5%) women had an EAS sphincter rupture before D2, whereas only 48 (15.7%) had a grade III OASI clinically recognized by the obstetrician in the delivery room during D1. These hidden ruptures were not accompanied by an increased risk of continence impairment after D2 (Table 2). The description of sphincter ruptures is provided Appendix E.

#### 3.5.2. Sphincter Tears after D2

A total of 143 (71.9%) women had sphincter ruptures revealed by the EAS performed after D2 (out of 199 who underwent EAS). Only six had “de novo” sphincter ruptures. These were women who had delivered vaginally, including one with episiotomy. The subgroup was too small to evidence a significant association between these ruptures and continence impairment in univariate analysis.

## 4. Discussion

### 4.1. Main Results

To our knowledge, this study is the largest prospective cohort describing risk factors for continence impairment after a second delivery in at-risk parturients. The main risk factor was instrumental extraction coupled with an episiotomy during a second vaginal delivery. These results are consistent with the literature and the absence of an increased risk related to episiotomy when analyzed independently is reassuring regarding the controversial impact of this procedure [7,24].

The risk factors of postpartum AI are still debated. Forceps delivery and perineal tears are frequently associated with AI [7,11,13,33]. But the association is less clear for episiotomy, multiparity [7,11], macrosomia and nulliparity [13]. In our work, none of these factors were associated with an increased risk of continence impairment after D2, possibly because we studied at D2 the impact of some events occurring at D1 and because we relied on a large and prospective cohort, unlike previous studies.

We found a strong association between the presence of urinary incontinence after D2 and the presence of AI. Patients and their physicians are often aware of the risk of urinary incontinence after delivery, while the diagnosis of AI is more challenging. The diagnosis of urinary incontinence could initiate dialogue between the obstetrician and the proctologist.

At inclusion, several women denied having AI during non-specific questioning, while a more focalized interview during the proctological consultation enabled a proper continence evaluation.

Surprisingly, the presence of AI before D2 was associated with a decreased risk of continence impairment after D2. We assume that some women had signs of AI related to the pregnancy itself, due to transit and pelvic static disorders, which improved after the delivery.

CS was not associated with a decreased risk of continence impairment after D2. This is consistent with the EPIC study [8] and recent literature [34,35].

### 4.2. The Place of EAS

Women with traumatic deliveries are routinely offered EAS examination in order to detect a clinically undiagnosed sphincter defect. We indeed observed that among half of the patients, an anal sphincter rupture had been missed clinically but was evidenced by EAS, which is consistent with the literature [6,9].

The EPIC study demonstrated however, that it was not beneficial to perform prophylactic cesarean section to these patients to prevent the occurrence of AI after D2 [8]. Our study goes further, showing that neither the presence of these hidden sphincter ruptures diagnosed before a second delivery, nor even the most severe sphincter tears, were associated with a significant risk of continence impairment after D2. We therefore have no argument for advocating the use of EAS after a first traumatic delivery. However, in our practice, performing an EAS, as frequently requested by maternities, allows these women at-risk to have access to a proctologist, who can inform them about long-term risk factors for AI and avoidable cofactors.

These results can be challenged by the relative scarcity of severe sphincter tears in our study, mostly because of the low percentage of internal sphincter injuries, which strongly impacts Starck’s score [29,30]. In Sultan’s cohort [9], 16% of women had an internal sphincter defect versus only 5.8% in our study. These results may be explained by the exclusion of patients with grade IV perineal tears or already reporting AI to their gynecologist after D1. There is therefore still room to debate the benefits of EAS among these patients.

### 4.3. Strengths and Limitations

We collected data in medical centers representative of the population pools of our region and we relied on validated clinical criteria (Vaizey score, MHU score) and paraclinical criteria (Starck’s score [29,30]). EAS were performed by a single expert operator, which limited ranking bias by preventing inter-operator variability. Moreover, the inter-operator concordance of this practitioner had already been validated in a previous work [7].

There is no consensus on thresholds defining AI in literature. We chose to define a continence impairment as an increase in two points of the Vaizey score after D2. This is a sensitive threshold for continence deterioration but not a very specific one.

Anal incontinence tends to decrease in the weeks following delivery and to reappear after menopause. One of the limitations of this work is the lack of follow-up beyond one year after D2.

Although the follow-up was prospective, we conducted this work after the legal closing date of the original study, which prevented us from completing the missing data. Therefore, 122/434 (28%) of women who underwent EAS before D2 were not analyzed. However, we did not evidence any difference between the baseline characteristics of our population and this unanalyzed population (Appendix F).

### 4.4. What Can Be Offered to a Woman at Risk of Continence Impairment after D2

Potential perineal damage risk can be reduced by measures such as regulating transit, treating dyschesia (source of pudendal stretch or prolapse) and avoiding additional sphincter trauma as much as possible. It is thus advisable to refer women at risk or presenting postpartum AI to proctologists for multi-disciplinary management.

Although the main risk factor for continence impairment is forceps delivery with episiotomy, these obstetrical interventions are sometimes necessary and cannot be completely avoided.

In case of proven AI and after managing possible cofactors, pelvic rehabilitation with biofeedback or neuromodulation of the sacral roots may be beneficial [36,37,38].

## 5. Conclusions

After a first traumatic delivery, instrumental delivery with episiotomy is the main risk factor for anal continence impairment during a second delivery. Women who present symptoms of urinary incontinence after their second delivery also are at greater risk of developing symptoms of anal incontinence. Referral to a proctologist will improve the detection and management of this incontinence. Caesarian section does not prevent it. In a woman wishing to start a second pregnancy after a first traumatic delivery, the presence of a sphincter lesion on endoanal ultrasound will not change the course of treatment.

## Figures and Tables

**Figure 1 jcm-12-01531-f001:**
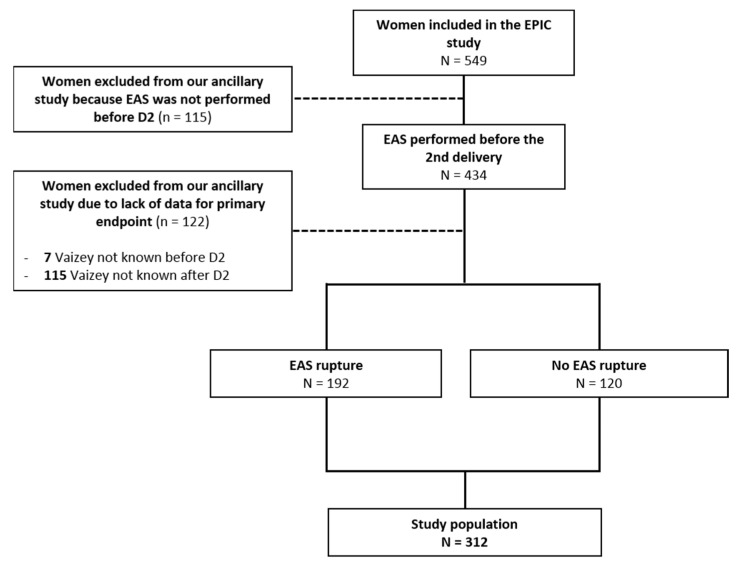
Flow Chart.

**Table 1 jcm-12-01531-t001:** Characteristics of the population.

	Total (N = 312)	NA
Median age [IQR]	33.4 [30.3–35.8]	0 5 5
Median BMI [IQR]	26.17 [23.9–29]	
Patients with BMI > 30	64 (20.8%)	
**Transit at baseline**		8
Constipation and/or Dyschesia	78 (25%)	
Diarrhea	8 (2.6%)	
Normal transit	211 (69.4%)	
**Geographical origin**		5
Asia	12 (3.9%)	
Europe	202 (65.8%)	
Maghreb	57 (18.6%)	
Other	36 (11.7%)	
**First delivery (D1)**		
Median gestational age in weeks [IQR]	40 [39–41]	5
Use of forceps	285 (92.2%)	3
Episiotomy	266 (86.9%)	6
Macrosomia	18 (5.8%)	4
Obstetrical anal sphincter injury (OASI)	95 (30.7%)	3
Grade I tear	27 (8.9%)	
Grade II tear	15 (4.9%)	
Grade III tear	48 (15.7%)	
EAS rupture after D1	192 (61.5%)	0
Hidden rupture after D1	146 (47.9%)	7
Severe rupture after D1 (Starck score ≥ 9)	18 (5.8%)	0
**Continence at baseline (before D2 visit)**		
Perineal rehabilitation performed after D1	225 (74.3%)	9
Median Vaizey score before D2 [IQR]	1 [0–3]	0
Transient anal incontinence	48 (15.7%)	7
**Detailed Vaizey score before D2**		0
Vaizey = 0	135 (43.3%)	
Vaizey = 1	52 (16.7%)	
Vaizey = 2	39 (12.5%)	
Vaizey = 3	22 (7.1%)	
Vaizey = 4	21 (6.7%)	
Vaizey ≥ 5	43 (13.8%)	
Median MHU score before D2 [IQR]	4 [2–8]	16
**Second delivery (D2)**		
Median gestational age in weeks [IQR]	39 [39–40]	87
Cesarean delivery	103 (35%)	16
Vaginal delivery	193 (65.2%)	16
Perineal rehabilitation performed after D2	201 (69%)	19

NA: Unavailable variables. IQR: Interquartile range. BMI: Body mass index. EAS rupture: Sphincter tear diagnosed in endoanal sonography. D1: First delivery, D2: Second delivery. MHU: Measurement of urinary handicap.

**Table 2 jcm-12-01531-t002:** Characteristics of the subgroup of women delivering vaginally during D2.

	Total (N = 193)	NA
Median age [IQR]	33.3 (30.5, 35.7)	2
Patients with BMI > 30	34 (18%)	
**Transit at baseline**		4
Constipation and/or Dyschesia	51 (27%)	
Diarrhea	5 (2.6%)	
Normal transit	132 (70%)	
**Geographical origin**		2
Europe	125 (65%)	
Maghreb	36 (19%)	
Other	30 (15.7%)	
**First delivery (D1)**		
Median gestational age in weeks [IQR]	40 [39.0–41.0]	18
Use of forceps	177 (93%)	2
Episiotomy	167 (88%)	2
Macrosomia	12 (6.3%)	2
Obstetrical anal sphincter injury (OASI)	49 (26%)	2
Grade I tear	18 (38%)	1
Grade II tear	7 (15%)	
Grade III tear	23 (48%)	
EAS rupture after D1	96 (50%)	3
Hidden rupture after D1	79 (42%)	
Severe rupture after D1 (Starck score ≥ 9)	7 (3.6%)	
**Continence at baseline (before D2 visit)**		
Perineal rehabilitation performed after D1	145 (77%)	5
Median Vaizey score before D2 [IQR]	1 [0–3]	0
Vaizey ≥ 5	25 (13%)	
Transient anal incontinence	31 (16%)	5
Median MHU score before D2 [IQR]	5 [2–7]	10
**Second delivery (D2)**		
Median gestational age in weeks [IQR]	40.0 [39.0–40.0]	79
Macrosomia	8 (6.1%)	62
Perineal rehabilitation performed	130 (71%)	9
Use of forceps	13 (6.8%)	1
Use of vacuum	12 (6.2%)	1
Median labor time [IQR]	4 [2–6]	68
Episiotomy	49 (26%)	2
Median Medio-lateral	2 (4%) 25 (51%)	22
Obstetrical anal sphincter injury (OASI)	114 (60%)	2
Grade I tear	87 (77%)	1
Grade II tear	24 (21%)	
Grade III tear	2 (1.8%)	
Grade IV tear	0 (0%)	

NA: Unavailable variables. IQR: Interquartile range. BMI: Body mass index. EAS rupture: Sphincter tear diagnosed in endoanal sonography. D1: First delivery, D2: Second delivery. MHU: Measurement of urinary handicap.

**Table 3 jcm-12-01531-t003:** Outcomes measured 6 months after D2.

General Population	Total (N = 312)	NA
Median Vaizey score after D2 [IQR].	1 [0–3]	0
Detailed Vaizey score after D2		
Vaizey = 0	137 (43.9%)	
Vaizey = 1	44 (14.1%)	
Vaizey = 2	40 (12.8%)	
Vaizey = 3	23 (7.4%)	
Vaizey = 4	14 (4.5%)	
Vaizey ≥ 5	54 (17.3%)	
Vaizey deterioration ≥ 2 points	67 (21.5%)	
Median MHU score after D2 [IQR]	1 [0–4]	13
EAS rupture after D2	143 (72%)	113
**Women delivering vaginally during the second delivery**	**Total (N = 193)**	**NA**
Median Vaizey score after D2 [IQR].	1 [0–3]	0
Detailed Vaizey score after D2		
Vaizey = 0	90 (47%)	
Vaizey = 1	30 (16%)	
Vaizey = 2	23 (12%)	
Vaizey = 3	17 (8.8%)	
Vaizey = 4	9 (4.7%)	
Vaizey ≥ 5	24 (12%)	
Vaizey deterioration ≥ 2 points	34 (18%)	
Median MHU score after D2 [IQR]	1 [0–4]	9
EAS rupture after D2	80 (66%)	71
“De novo” sphincter rupture †	6 (3.6%)	28

NA: Unavailable variables. IQR: Interquartile range. EAS rupture: Sphincter tear diagnosed in endoanal sonography. D1: First delivery. D2: Second delivery. MHU: Measurement of urinary handicap. †: Rupture considered “de novo” if no sphincter rupture was evidenced in EAS before D2.

**Table 4 jcm-12-01531-t004:** Factors associated with a worsening of the Vaizey score ≥ 2 points after second delivery (Univariate analysis).

Total Population (N = 312)	Cases (N = 67)	OR (95% CI)	*p* (Wald)
**General parameters**			
Obesity (BMI ≥ 30)	17	1.43 (0.76–2.71)	0.269
Maghreb ethnicity	14	1.24 (0.63–2.44)	0.533
Transient anal incontinence after D1 (<2 months)	10	0.99 (0.46–2.11)	0.978
Normal transit before D2 (vs abnormal) (n = 211)	44	0.9 (0.5–1.63)	0.735
Perineal rehabilitation performed after D1	47	0.95 (0.51–1.77)	0.866
Perineal rehabilitation performed after D2	40	0.92 (0.48–1.75)	0.789
MHU score before D2 (per point)		1.06 (0.99–1.12)	0.098
MHU score after D2 (per point)		1.16 (1.07–1.26)	<0.001
D2 by caesarean section	30	1.92 (1.09–3.38)	0.023
Anal incontinence (Vaizey ≥ 5) before D2	4	0.34 (0.12–0.97)	0.045
**Obstetrical parameters related to the 1st delivery**			
Macrosomia D1	5	1.44 (0.5–4.21)	0.501
Use of forceps D1	60	0.8 (0.3–2.1)	0.651
Episiotomy D1	54	0.59 (0.28–1.24)	0.168
**Analysis of sphincter ruptures**			
Sphincter rupture diagnosed in EAS before D2:			
Absence		Reference	
Presence	46	1.49 (0.84–2.64)	0.178
Hidden rupture before D2 status:			
Absence of EAS sphincter rupture	18	Reference	
Hidden rupture diagnosed	34	1.57 (0.83–2.96)	0.164
Grade III OASI	14	2.13 (0.95–4.74)	0.065
Severity revealed in EAS before D2:			
Absence of EAS sphincter rupture	21	Reference	
Mild sphincter rupture Severe sphincter rupture (Starck ≥ 9)	42 4	1.5 (0.84–2.69) 1.35 (0.4–4.5)	0.174 0.629

BMI: Body mass index. EAS rupture: Sphincter tear diagnosed in endoanal sonography. D1: 1st delivery. D2: 2nd delivery. MHU: Measure of urinary handicap, analyzed as a continuous variable. OASI: Obstetrical anal sphincter injury.

**Table 5 jcm-12-01531-t005:** Factors associated with a worsening of the Vaizey score ≥ 2 points after second delivery (Multivariate analysis).

Total Population (N = 312)	OR (95% CI)	*p* (Wald)
D2 by cesarean section	1.56 (0.79–3.08)	0.201
Anal incontinence (Vaizey ≥ 5) before D2	0.27 (0.08–0.86)	0.027
MHU score after D2 (per point)	1.22 (1.11–1.33)	<0.001
Hidden rupture (versus no EAS rupture) Grade III-IV OASI (versus no EAS rupture)	1.26 (0.59–2.72) 2.18 (0.83–5.73)	0.554 0.113

D2: Second delivery EAS rupture: Sphincter tear diagnosed in endoanal sonography. OASI: Obstetrical anal sphincter injury. MHU: Measure of urinary handicap, analyzed as a continuous variable.

**Table 6 jcm-12-01531-t006:** Factors associated with a worsening of the Vaizey score ≥ 2 points after second delivery in subgroup of women delivering vaginally (Univariate analysis).

Population (N = 193)	Cases (N = 34)	OR (95% CI)	*p* (Wald)
**General parameters**			
Obesity (BMI ≥ 30)	8	1.62 (0.6–6.4)	0.291
Maghreb ethnicity	8	1.49 (0.61–3.64)	0.386
Transient anal incontinence after D1 (<2 months)	5	0.97 (0.34–2.76)	0.953
Normal transit before D2 (vs Abnormal)	21	0.79 (0.35–1.77)	0.569
Perineal rehabilitation performed after D1	26	1.66 (0.6–4.63)	0.332
Perineal rehabilitation performed after D2	22	0.80 (0.36–1.78)	0.579
MHU score before D2 (per point)		1.11 (1.01–1.22)	0.031
MHU score after D2 (per point)		1.21 (1.08–1.37)	0.001
Anal incontinence (Vaizey ≥ 5) before D2	2	0.37 (0.08–1.65)	0.192
**Obstetrical parameters related to the 1st delivery**			
Macrosomia D1	2	0.95 (0.2–4.57)	0.954
Use of forceps D1	30	0.75 (0.2–2.85)	0.67
Episiotomy D1	27	0.51 (0.18–1.43)	0.203
**Analysis of sphincter ruptures**			
EAS rupture before D2	17	1.01 (0.48–2.12)	0.973
Hidden rupture before D2 status:			
Absence of EAS rupture	14	Ref	
Hidden rupture before D2	14	1.14 (0.51–2.56)	0.754
Grade III perineal tear before D2	5	1.47 (0.47–4.61)	0.51
Severity revealed in EAS before D2:			
Absence of EAS sphincter rupture	17	Ref	
Mild sphincter rupture	16	1.03 (0.49–2.19)	0.936
Severe EAS sphincter rupture (Starck ≥ 9)	1	0.78 (0.09–6.94)	0.827
EAS rupture after D2:			
Absence of sphincter rupture	6	Ref	
“De novo”	2	2.25(0.33–15.26)	0.406
Worsening of previous lesions	4	1.13 (0.28–4.6)	0.87
Stability of previous lesions	19	0.98 (0.36–2.71)	0.973
**Obstetrical parameters related to the 2nd delivery**			
Macrosomia D2	1	0.59 (0.07–5.02)	0.628
Instrumental delivery at D2:			
Non instrumental	26	Ref	
Instrumental: Forceps or vacuum	8	2.73 (1.06–7.03)	**0.037**
Episiotomy D2 (n = 49)	12	1.77 (0.8–3.91)	0.159
Instrument/Episiotomy combination:			
No Instrument and no episiotomy at D2	20	Ref	
D2 with instruments/without episiotomy	2	1.22 (0.25–6.08)	0.806
D2 with episiotomy/without instrument	6	1.06 (0.39–2.88)	0.902
D2 with instruments and episiotomy	6	5.5 (1.61–18.78)	**0.007**

BMI: Body mass index. D1: 1st delivery. D2: 2nd delivery. EAS rupture: Sphincter tear diagnosed in endoanal sonography. MHU: Measure of urinary handicap, analyzed as a continuous variable.

**Table 7 jcm-12-01531-t007:** Factors associated with a worsening of the Vaizey score ≥ 2 points after the second delivery in the subgroup of women delivering vaginally (Multivariate analysis).

Population (N = 193)	OR (95% CI)	*P* (Wald)
Anal incontinence (Vaizey ≥ 5) before D2	0.22 (0.04–1.14)	0.071
MHU score after D2 (per point)	1.25 (1.1–1.43)	<0.001
Instrument/Episiotomy combination at D2:		
No Instrument and no episiotomy at D2	Ref	
D2 with instruments/without episiotomy	1.34 (0.26.6.91)	0.729
D2 with episiotomy/without instrument	1.03 (0.34.3.12)	0.953
D2 with instruments and episiotomy	4.18 (1.05–16.59)	0.042

D2: 2nd delivery. MHU: Measure of urinary handicap, analyzed as a continuous variable.

## Data Availability

All data are available on request to the corresponding author.

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
