# Peer review of "Risk Factors for Anal Continence Impairment Following a Second Delivery after a First Traumatic Delivery: A Prospective Cohort Study"

_jcm, 2023, doi:10.3390/jcm12041531_

Round 1
Reviewer 1 Report
Thank you for requesting to provide a review of this article, which has a subject of high interest.
The main purpose of the analysis was to study the risk factors for anal continence impairment following the second delivery.
The main question adressed in the research was whether the presence of urinary incontinence and the combined use of instruments and episiotomy during the second delivery is the main risk factor for anal continence impairment. Also, the main objectives of the study were to establish the risk factors for this kind of pathology after second delivery, among women who had a traumatic first delivery and, in particular, the impact of endoanal sonography ruptures.
The article is an ancillary study of the prospective, randomized, multicentered analysis, following a number of 312 women with a history of traumatic first delivery, who were consulted before and 6 months after the second delivery, for a period of time between April 2008 and December 2014. The topic is original and relevant in the field and brings usefull knowledge regarding the subject. A comprehensive search strategy was used and so, it was demonstrated that the main risk factor for anal continence impairment is the instrumental extraction, coupled with an episiotomy during a second vaginal delivery. The review methodology was comprehensive with screening and data extraction. When it comes to the methodology used, no specific improvements should be considered from my point of view.
The conclusions are consistent with the evidence and the arguments presented, and they adress properly to the main question which conducted the analysis.
The references are appropriate and well suited for this kind of study.
Regarding the figures and pictures used in the article, they provide suitable information about the cases and show significant statistical references. They are also well understandable and the information is easy to be followed. There are no other comments required about these items, from my point of view.
Regarding the structure and accuracy of the phrases, the manuscript has well structured information, with supported evidence and well structured phrases.
The manuscript is original and well defined. The results provide an advance in current knowledge. The results are being interpreted appropriately and are significant, as well as the conclusions.
The article is written in an appropriate way.
The study is correctly designed and the analysis is being performed at high standards, so the data are robust enough to draw the conclusion.
Surely the paper will attract a wide readership.
The English language is appropriate and well understandable.
There are a few things to add in the lines below, but the article should be published after the corrections are made:
Line 68: „,” before „in particular” and after „in particular”
Line 103: were retained, not „have been retained”
Line 151: have delivered, not „had delivered”
Author Response
Dear colleague.
Thank you for taking the time to review this manuscript and thank you for your favorable feedback.
I will make the corrections you advised.
Sincerely
Gabriel Marcellier
Reviewer 2 Report
This is an important and well carried out study on an important topic, incontinence after a second vaginal delivery where the first entailed perineal trauma. Although the results are not altogether new or surprising, they confirm some earlier results. The study was done in a prospective way and that adds credibility to a considerable degree. The study is a sub-study from a larger prospective trial, the EPIC study, but that is fully acceptable.
The introduction is well presented and then information is given in good and fully acceptable detail. The statistical analysis is relevant and the Discussion balanced. It is interesting that among the nearly 200 women who delivered their second child vaginally after a previous vaginal traumatic delivery, the presence of a rather large baby, a sphincter defect seen on endoanal ultrasound and perineal exercises did not matter so much. It was the need for a new instrumental delivery, which usually will be accompanied by episiotomy (?again), that mattered to produce a worsening of anal incontinence. Anal incontinence is a severe problem, but has remained a hidden health topic. So here this study adds importantly to knowledge. More research will be needed on the situation of what to do next after a traumatic vaginal delivery, both for obstetricians/midwives and for proctologic surgeons, in order to define what can be done to forestall such an occurrence in this specific group, as well as repair damage. It is hard to predict who these women will be and there is little one can do as an obstetrician once an instrumental delivery is needed again. But it is then that adequate repair techniques, attention to detail and experience is needed.
The scientific English is good in general, but can be improved in many places, including the use of prepositions, and attention to singular/plural word forms. In a scientific article colloquial shortening such as wasn´t is not used and it is the Royal College of Obstetricians and Gynaecologists. EAS does not estimate (p. 2, line 56), something is estimated by EAS. The authors could explain better what is meant by perineal rehabilitation.
Author Response
Dear colleague.
Thank you for taking the time to review this manuscript and thank you for your favorable feedback.
I will make the corrections regarding the English.
As for the perineal rehabilitation, the patients were simply asked if they had done it or not, the number of sessions and the method were not collected. The latter were heterogeneous in our cohort (manual or with probe) but I do not have any statistics on this subject. I will clarify this in the manuscript.
Sincerely
Gabriel Marcellier